# Activating beyond Informing: Action-Oriented Utilization of WeChat by Chinese Environmental NGOs

**DOI:** 10.3390/ijerph19073776

**Published:** 2022-03-22

**Authors:** Jing Xu, Huijun Zhang

**Affiliations:** 1Department of Sociology, School of Humanities and Social Science, Xi’an Jiaotong University, Xi’an 710049, China; 2Department of Physics, The Hong Kong University of Science and Technology, Clear Water Bay, Hong Kong, China; huijun.zhang@hotmail.com

**Keywords:** social media, WeChat, environmental NGOs, China

## Abstract

Social media has generated new opportunities for non-governmental organizations (NGOs) to inform and educate publics, and more powerfully, it enables NGOs to mobilize people to act. To enhance understanding how social media functions to serve action-oriented communication for organizations, we focused on WeChat, the largest social media in China. We examined an original dataset of 2472 articles posted by 175 environmental NGOs (ENGOs) during a two-month period in 2019 on WeChat. A qualitative content analysis was conducted to identify what actions ENGOs use WeChat to mobilize. We used statistical methods to analyze the effectiveness of ENGOs’ action-oriented utilization of WeChat and the organizational differences in the adoption of action-oriented messages. This study reveals that more than a quarter of the articles were mobilizational in nature. There were three major types of action ENGOs mobilize via WeChat. Though the informational use of WeChat is extensive, users prefer action-oriented messages and the activating strategy is more effective in motivating users to engage. Our analysis indicates that the more participatory people feel, the more likely they are to respond to WeChat messages, and the more they are involved. Our findings have implications for how the adoption of WeChat helps engender new paradigms of citizen participation.

## 1. Introduction

The advent of social media has ushered in a new era of opportunities for organizations to communicate and engage with their community. Among the most active social media users are non-governmental organizations (NGOs), which use social media messaging as a central communicative tactic. Social media helps NGOs disseminate and share information more extensively and effectively to a vast audience, and more powerfully, it enables NGOs to mobilize their publics to do something concrete, either donating money, attending an event, volunteering, or any other activities that relate to fulfilling their missions [1]. As one of the key characteristics of social media lies in interactivity and decentralized engagement, NGOs can tap into the mobilization potential of the platform to reach out to a vast network of members, donors, volunteers, supporters, and the community at large, and actively involve them. Social media represents a significant source of opportunity for NGOs to enhance citizen participation and active involvement. The mobilizational use of social media helps serve practical everyday purposes, such as raising funds, increasing volunteer support, and recruiting partners [2]. This may be what many organizations ultimately want to achieve. However, to date, few studies have explored whether and how NGOs mobilize action on social media and the effectiveness of action-oriented messages.

This study represents a focused effort to address this gap by examining the action-oriented communication of environmental NGOs (ENGOs) on WeChat in China. The development of Chinese ENGOs and the advancement of information and communication technologies (ICTs) have been closely embedded or mutually constitutive [3]. Since the 1990s, the Internet has empowered resource-poor ENGOs concerning self-representation, information dissemination, network building, and environmental collective actions [4]. With the advancement of ICTs to the evolutionary stage, commonly described as Web 2.0, social media has gained popularity among ENGOs in the country. Few studies have found that the social media employment of ENGOs in China can contribute to enhancing their communicative capacity and facilitating citizen engagement [5,6,7]. Yet, those studies with a heavy focus on a specific ENGO or a specific case study have barely touched on a message-level analysis of the content of communications by ENGOs on social media. Moreover, prior research has predominantly focused on Weibo, a leading microblogging platform in China. Little attention has been paid to WeChat—China’s largest and fastest-growing social media platform—which is probably because it is much more difficult to scrape content from WeChat than from other social media platforms. ENGOs rely heavily on their supporters and constituents to achieve the objectives of protecting the environment; effective leverage of communication channels is pivotal to their strategic success. China is a highly Internet-connected country; almost everyone has a mobile phone and almost everyone is on WeChat. According to the latest statistics, in December 2021, there were over 1.64 billion mobile phone users in China, where the population is around 1.42 billion [8]. WeChat is the most popular application in the country, its monthly active users reached over 1.26 billion by the end of 2021 [9]. Thus, it has great mobilization potential for NGOs to capitalize on. This implies that WeChat provides a large base of potential supporters, donors, volunteers, and activists for ENGOs. ENGOs should, therefore, take advantage of the potential for citizen participation that WeChat offers. It is important to know and understand which forms of action-oriented communications on WeChat are employed and most effective and best serve the organizational mission.

Drawing upon the detailed data of the WeChat official accounts of 175 ENGOs for two months in 2019, we conducted the first comprehensive study of ENGOs’ action-oriented communication on WeChat. In this study, we focus on two levels of analysis: organization and message. At the message level, the study aims to examine (1) How do ENGOs in China use WeChat to mobilize action? (2) How effective is the action-oriented utilization of WeChat by ENGOs? At the organizational level, it explores (3) What are the characteristics of ENGOs that adopt action-oriented communication? The rest of this study is structured as follows: a literature review and presentation of the theoretical foundation of the study; data and methodology; results; and discussion of our findings. We conclude with theoretical and practical implications and suggest possible directions for future research.

## 2. Literature Review

### 2.1. The Social Media Platform—WeChat

Mainstream social media platforms, such as Facebook and Twitter, are not accessible in Mainland China. Instead, domestic social media, such as WeChat and Weibo have flourished. Launched by Tencent in 2011, WeChat has become the most widely used social media platform in China. WeChat is a smartphone-based mobile application that provides multiple functions, including instant text and voice messaging, content provision, social networking, and payment [10]. WeChat enables one-to-one or small-group communication and allows brands, organizations, and news outlets to establish official accounts that provide content and updates to an unlimited number of followers [11]. More than 20 million organizations have opened official accounts on WeChat to interact and engage with people. The principal manner of message dissemination for WeChat official accounts is through the publishing of articles.

WeChat official accounts are employed for broad-based communication, a service that functions much the same as Weibo [12]. In contrast to Weibo, where messages are restricted to 140 characters, there is no limit to the number of characters in articles by official accounts on WeChat. However, there is a limit to the posting frequency of WeChat official accounts. Subscription accounts, which are a major type of WeChat official account, are only allowed to post content once a day, with a maximum of eight articles per post. WeChat official accounts are intended to provide high-quality content that is informative, useful, unique, and interesting enough to attract, engage, and retain followers, as opposed to a high frequency of posts that create a poor user experience [13]. WeChat, thus, is seen as a suitable platform for posting long, detailed, insightful, and text-based articles with the inclusion of rich media content. Compared with microblogging, WeChat is more private and exclusive, more relationship-focused and less visible, and this may make it harder to gather a large number of followers. One cannot simply “follow” any account of an individual or organization, as on Weibo, but must request permission to be in a specific circle. The higher the threshold for entry, the tighter the community within each user’s network becomes, which means that there is a stronger sense of community on WeChat than on Weibo [14].

### 2.2. Communication Strategy

To survive, all types of organizations need to inform their publics that their organizations’ activities are desirable, proper, or appropriate within certain social norms and values. Particularly, for NGOs to achieve their missions they must rely on public support—the larger and more dedicated their publics, the better equipped these organizations are to raise funds, recruit volunteers, and promote action [15]. Social media is a powerful, convenient, and affordable tool for both of these ends. Drawing from extant literature, two communication strategies adopted by NGOs on social media are examined: informing and activating. The informing strategy and the activating strategy are compatible with two communication models of public relations by Grunig and Hunt: public information and two-way asymmetry [16]. First, NGOs use social media to impart unidirectional messages to audiences and to inform their community, which helps NGOs enhance organizational visibility and legitimacy, raise awareness, foster relationships with stakeholders, and build trust [17,18,19]. There are three issues the informing strategy should address: (1) spreading information related to an organization, its mission, programs, accomplishments, or other relevant information of potential interest to its publics, (2) providing useful content, such as knowledge, ideas, and viewpoints to attract and educate publics; (3) sharing something that strengthens ties to stakeholders, such as highlighting their stories in the NGOs’ posts [20,21,22,23,24,25].

Alternatively, a focus of the activating strategy is to encourage and empower the public to do something for or on behalf of an organization—anything from making donations or participating in online and offline activities to becoming volunteers, interns, or employees [1,6]. It involves the promotional and mobilizational uses of social media messages, where, implicitly, at least, users can be regarded as a valuable resource that can be mobilized to help the organization promote its causes. The ultimate goal of many organizations is often to mobilize their online networks to do something concrete to help achieve their goals [26]. Action-oriented messages serve as a tool to involve the public, move people’s attitudes toward the organization, and lead to supportive behavioral results, such as volunteering, donating, or advocating for the organization [2]. Therefore, the activating strategy is perhaps the most tangible, outcome-oriented approach for engaging the public on social media. Guo and Saxton developed a three-stage hierarchical framework for social-media-based NGO advocacy, including reaching out to people, keeping the flame alive, and stepping up to action [21]. Asker and Smith proposed a four-wing framework—focus, grab attention, engage, and take action—for organizations to fully tap into the function of social media [27]. These findings revealed that action mobilization represents the highest goal of social media engagement.

Organizations pursuing the informing strategy try to “give sense” to their publics. However, the activating strategy allows NGOs to inspire and motivate social media users to act, thereby moving their publics, in effect, from informed individuals to a community of donors, volunteers, activists, and partners [22]. This type of public engagement, which can develop into offline support, mobilization, and activism, differs substantially from the one based on the informing strategy [28]. The actions NGOs get the public to take may reflect what NGO leaders perceive as the greatest value they can derive from social media communication [29]. Based on the importance of action mobilization on social media, the following research questions were proposed:

**RQ1**: What actions do ENGOs in China use WeChat to mobilize?

**RQ2:** To what extent do ENGOs communicate action-oriented messages on WeChat?

### 2.3. Public Engagement on WeChat

It is through articles that organizations on WeChat reach out to and engage with people to help achieve mission objectives, such as mobilizing constituents to collective actions. The achievement of these objectives is contingent on first persuading the public to read their articles. WeChat users can gain access to the articles posted by the WeChat official accounts they follow and the articles their WeChat friends share on the platform. Thus, WeChat articles reach a broad readership far beyond the followers of official accounts. The number of *Reads* of an article is open to the public. Similar to other social media platforms, WeChat also offers engagement tools, such as *Shares* and *Comments.* However, because of the strict rules for WeChat, the number of *Shares* of an article is not available to the public; comments on WeChat articles are not fully accessible because WeChat only displays the selected comments. Therefore, *Reads* is the primary indicator used to measure the level of engagement on WeChat.

Engagement is defined as “the interaction between an organization and those individuals and groups that are impacted by, or influence, the organization” [30]. Because social media are relationship-centric and inherently participatory, engagement with social media messaging can contribute to the cultivation of relationships, supportive behaviors, and meaningful involvement. There are two factors that could affect public engagement on WeChat. First, communication strategy may result in a change in public engagement. Tsai and Men proposed public engagement as a behavioral construct with hierarchical activity levels and classified engagement and its associated activities into two levels: reactive consuming and proactive contributing [31]. This falls in line with the informing strategy and the activating strategy, which contribute to passive message consumption and active two-way interaction, respectively [32]. We posit that action-oriented messages elicit more engagement from the public than messages based on the informing strategy. Supporting evidence can be found in the literature. For example, in a study of NGOs’ Facebook messages, Saxton and Waters employed the number of likes, shares, and comments to evaluate different levels of stakeholder engagement and found that call-to-action messages, such as encouraging volunteering, are viewed more favorably and elicit more likes and comments than informational messages [1]. Guided by the literature, we propose the following hypothesis:

**H1:** 
*Publics are more likely to read action-oriented articles than non-action-oriented articles.*


Second, the type of action may also affect public engagement. Public participation in environmental governance has been on the rise in China during the past two decades. The heightened public awareness of environmental degradation and mounting health concerns has driven people to become involved in environmental issues [33]. Meanwhile, post-materialistic values, such as protecting the environment, have become increasingly prevalent in Chinese society recently [7]. Indeed, China has been experiencing a blossoming of environmental awareness and pro-environmental behaviors among the public. According to a recent survey, more than 80% of Chinese urban residents favor doing something for environmental protection, and 82.7% want to volunteer to help protect the environment [34]. Furthermore, the Chinese political and social climate has been gradually evolving toward a participatory approach to environmental governance, leading to much more space for the public to become involved [35]. Topics related to environmental volunteers/interns/staff members demonstrate an immediate opportunity that people can harness to work for the environment. Given this discussion, we propose the following hypothesis:

**H2:** 
*Publics are more likely to read articles related to environmental volunteering/internships/jobs than other types of action-oriented articles.*


### 2.4. ENGOs in China

Since the 1990s, China has seen rapid development of ENGOs in number and in function and social impact [36]. It is widely recognized that ENGOs have been the oldest, most dynamic, and most influential NGOs in China [37]. Some researchers have argued that the emergence and growth of Chinese ENGOs can be attributed to the combination of favorable political conditions and the opportunities offered by the media, the Internet, and international NGOs [38]. ENGOs have played an active and important role in a wide array of environmental issues—ranging from ecological education to wildlife conservation to pollution control. Chinese ENGOs have long embraced the idea of harnessing the power of ICT to advance their missions. They have been early adopters of the Internet and active users of social media [3]. Most notably, Chinese ENGOs have recently begun to apply large numbers of data to improve public participation, leading to the creation of green social capital in China [39]. Regarding organizational function, ENGOs in China can be divided into three types: (1) solution-focused ENGOs that aim to implement on-the-ground projects and programs to help address a specific environmental issue; (2) service-focused ENGOs that seek to engage people with nature and cultivate environmental literacy and stewardship; and (3) supportive ENGOs that provide funding, knowledge, expertise, ideas, and other resources to promote environmental protection [40].

The development of ENGOs in China varies by location. Beijing, China’s capital and political center, allows NGOs to solicit larger funds, obtain greater resources, and develop more significant connections [41]. Hence, in Beijing, the ENGO sector has grown the largest and arguably the highest level of maturity in the country [42]. Yunnan province, which is characterized by huge underdeveloped areas and rich biodiversity, is home to a vibrant ENGO community [43]. Guangdong province and the municipality of Shanghai, which are known for their economic prosperity and openness to international influences, also have a vibrant ENGO sector, while, in many other regions, ENGOs exist on a very limited scale and impact [42]. The third question was proposed as follows:

**RQ3**: How does ENGO action-oriented communication on WeChat differ by organizational type and location?

We posit that characteristics of an organization’s social media network, particularly its size, affect whether and to what extent the organization uses social media messages to mobilize the publics to act. There is much intuitive sense in this assertion, and a growing line of research has found evidence that social networks provide potential supporters, donors, and volunteers that NGOs can tap into. For example, Eng et al. found that NGOs’ resource acquisition, such as volunteer recruitment, is strongly related to relationally embedded networks ties [44]. As suggested by Reddick and Ponomariov, higher levels of involvement in voluntary associations (for example, through membership or some other form of participation) are more likely to result in a higher inclination to effect citizen participation, such as online donating [45]. The size of the audience on social media platforms is a huge resource on which NGOs can capitalize to enhance their capacities and missions [17]. It seems reasonable to expect that the larger their publics, the more motivated organizations are to take advantage of the resources, and the more likely organizations are to mobilize the public to take action. This leads us to the following hypothesis:

**H3:** 
*ENGOs that communicate action-oriented messages have a larger social network size than those that do not.*


## 3. Methods

### Sampling and Data Collection

The sample for this study was drawn from the first Chinese ENGO map developed by the Heyi Institute, an independent research-oriented institution and a leading data source for ENGOs in the country. This study only included NGOs with a focus on environmental protection, that is, NGOs for which the primary goals and activities are to improve the environment. Moreover, Chinese NGOs are divided into two types. The first type is “government-organized NGOs” that are supported and sponsored by the Chinese government. The second type is grassroots NGOs that are initiated by private citizens and are similar to their Western counterparts. This study focused solely on grassroots ENGOs, government-organized ENGOs were excluded from the sample. After these criteria were applied, 358 active ENGOs across the country were identified. To determine which organizations had WeChat official accounts, we first used the WeChat search function for the names (or abbreviated names) of the 358 ENGOs. Most of the organizations’ WeChat official accounts were identified through this process. A few provided a QR code on their websites that links directly to their WeChat official accounts, while others were identified by searching Google or Baidu, the major search engine in China. Among the 358 ENGOs, 281 were found to have WeChat official accounts.

We chose to gather WeChat utilization data on these organizations from 1 August 2019 to 30 September 2019. ENGOs are more active during this period, when they are actively engaged in environmental education programs, citizen participation projects, volunteer/internship activities, and donation campaigns. ENGOs thus need to use WeChat more heavily to mobilize people to take action. The data collection period chosen represents a good opportunity to examine ENGOs’ action-oriented utilization of WeChat, allowing us to better understand the action-oriented communicative function WeChat serves for the organizations.

Among the 281 ENGOs, 106 organizations did not post updates during the data collection period. These organizations were deemed inactive and screened out. Hence, 175 ENGOs were finally included in the study’s sample, which was from 29 different provinces/autonomous regions/municipalities. WeChat application programming interface is strictly restricted, making it difficult if not impossible to download WeChat posts with software. Therefore, all organizational posts published during the period were downloaded by one of the authors manually, and 2472 WeChat articles were finally collected.

## 4. Measures

According to NGO communication strategies, ENGOs’ WeChat messages can be classified into two categories: action-oriented messages following the activating strategy, non-action-oriented messages involving the informing strategy. The coding scheme was developed through a bifurcated process. Based on existing research on NGO and social media, we first studied previously identified strategies for informing and activating the public in the existing literature [1,22,25]. We then conducted a qualitative inductive analysis to identify new strategies adopted by ENGOs to inform their publics and promote action in the context of WeChat and, thus, are previously unseen in the literature, leading to a series of new codes. This approach is consistent with the methodological literature in which qualitative inductive analysis is viewed as more appropriate for grounded theory building [46].

We modified five previously identified codes for application to this case: (1) two codes for action-oriented messages, including “working with ENGOs as interns/volunteers/employees” and “donating”; (2) three codes for non-action-oriented messages, including “ENGO activities and performance”, “sharing stories of stakeholders”, and “holiday greetings and giving thanks”. Furthermore, a new activating strategy used by ENGOs was identified and coded as “registering to participate in an activity” for action-oriented messages, which involves seven sub-codes. We also identified a new informing strategy and created a new code “environmental knowledge and policy” for ENGOs’ non-action-oriented messages. As a result, a total of seven codes were developed in this process, three for ENGOs’ action-oriented messages, the other four for their non-action-oriented messages.

The effectiveness of the action-oriented utilization of WeChat by ENGOs was evaluated through the level of public engagement in ENGOs’ messages. In order to assess whether action-oriented utilization of WeChat is more effective to elicit engagement, we compared the level of WeChat user engagement in ENGOs’ action-oriented messages and non-action-oriented messages. As discussed above, *Reads* is the primary indicator used to measure the level of engagement on WeChat. Therefore, the number of reads in each article was coded.

ENGO action-oriented communication was defined by whether an ENGO communicated action-oriented articles in the two-month period; it was coded as 1 (yes), and 0 (no). ENGO organizational types were coded into three categories: solution-focused, service-focused, and supportive ENGOs. Coding categories were based on mission statements on ENGOs’ WeChat official accounts. Most ENGOs placed their mission statements on the front page of their accounts on WeChat, and some of them had mission statements on their websites or publications, such as annual reports. ENGO locations referred to the province or municipality of the ENGO’s headquarters based on the self-reporting information shown in their WeChat profiles, and were coded by two categories regarding ENGO development by province. As suggested by the literature, Beijing, Shanghai, Guangdong, and Yunnan are classified into regions of high-level ENGO development, and all other provinces are classified under regions of low-level ENGO development. Social network size was measured by the followers of an ENGO’s WeChat official account. To evaluate the relationship between social network size and action-oriented communication, the followers of each ENGO’s WeChat official account were coded.

The two authors conducted the coding process. We used an iterative process of pilot testing and refining the coding rules before starting with the actual coding. The initial coding scheme for ENGO communication strategies consisted of a total of eleven codes: four for ENGOs’ action-oriented messages and seven for non-action-oriented messages. Using this coding scheme, we first coded 100 articles tentatively and found that there was substantial overlap, so we collapsed similar items into a single code. The revised coding scheme was developed and then used to code the 100 articles again. Discrepancies between coding were discussed and coding rules refined until 100% agreement was reached. As a result, the refined coding scheme with seven codes was built and another 230 articles were coded. The codes were exhaustive and mutually exclusive. Each article was assigned a single code from the new coding scheme. Intercoder reliability was assessed by the two authors based on the 230 articles with 95% intercoder agreement and a Cohen’s kappa score of 0.94, which indicates a high level of reliability.

## 5. Results

To answer RQ1 about what actions ENGOs utilize WeChat to mobilize, a frequency analysis was conducted (*n* = 654). Table 1 shows that ENGOs used WeChat to mobilize people to take three types of actions: registering for an environmental activity, working with ENGOs as a volunteer/intern/staff member, and donating. ENGOs most often mobilized people to register for various environmental activities (61.6%). ENGOs’ articles provided detailed information about an activity, the date, time, location, price, and ways to attend it and often included a QR code that users could use to easily register and participate. ENGOs also posted internship/volunteering/job opportunities on WeChat to motivate people to work with them for the environment (22.3%). To encourage people to become involved, ENGOs’ articles often highlighted their missions and what interns/volunteers/staff members can do to help organizations and protect the environment. Moreover, WeChat was also employed by ENGOs to mobilize people to donate (16.1%). ENGOs posted donation appeals to involve the public in a fundraising program called 99 Giving Day. Because of its easy payment function, WeChat is a convenient way for people to make donations online. To inspire the public to act, ENGOs’ articles often underscored what they have done for environmental protection and asked for donations to help them keep moving forward.

To answer RQ2, which addressed the extent to which ENGOs communicate action-oriented messages on WeChat, we carried out a frequency analysis (*n* = 2472). Table 2 shows that 654 action-oriented articles were communicated in the two-month period, which accounted for 26.5% of all articles. Meanwhile, ENGOs also communicated 1818 non-action-oriented articles, and most of them were employed to update the public about ENGOs’ most recent activities and highlight their achievements (53.9%). WeChat also served as a platform for ENGOs to inform and educate the public about environmental knowledge and policy (26.6%). The other non-action-oriented articles (19.5%) were used to share the stories of stakeholders, such as volunteers, and to send holiday greetings and messages of thanks to their followers.

To test H1, which explored different levels of engagement (number of reads) between action-oriented articles and non-action-oriented articles, a *t* test was performed. Table 3 presents the descriptive information of the reads of ENGOs’ articles. Action-oriented articles had more reads (*M* = 555.50, *SD* = 894.670) than non-action-oriented articles (*M* = 354.82, *SD* = 862.890, *t* = 4.987, *p* < 0.001). Publics were significantly more likely to read action-oriented articles than non-action-oriented articles. Therefore, H1 was supported.

H2 explored which type of action reflected on ENGOs’ WeChat articles elicited more readership from publics. First, the qualitative content analysis indicated that there were three major types of actions ENGOs mobilized on WeChat: becoming a volunteer/intern/staff member, registering to participate in an environmental activity, and donating. A one-way ANOVA test suggests that public engagement levels significantly varied by the type of action (*F* (2, 651) = 3.764, *p* < 0.05). Post hoc LSD analysis revealed that publics were significantly more likely to read articles related to environmental volunteering/internships/jobs (*p* < 0.05). There was no statistically significant difference between donation appeal messages and environmental activity articles (*p* = 0.911). Therefore, H2 was supported.

To answer RQ3 about whether ENGO action-oriented communication on WeChat differs by location and organizational type, we found that among the 175 ENGOs, 119 (68%) communicated action-oriented messages, but 56 (32%) did not send action-oriented messages in the two-month period. There were 63 ENGOs (36%) based in the region of high-level ENGO development, including Beijing, Shanghai, Guangdong, and Yunnan; 112 ENGOs (64%) were in other regions of low-level ENGO development. A chi-square test demonstrated that ENGOs in the region of high-level ENGO development (77.8%) were significantly more likely to communicate action-oriented messages on WeChat: (χ^2^ (1, *n* = 175) = 4.325, *p* < 0.05). Regarding organizational type, a chi-square test revealed that serviced-focused ENGOs (85.7%) were significantly more likely to communicate action-oriented messages on WeChat: (χ^2^ (2, *n* = 175) = 6.315, *p* < 0.05).

To address H3, which predicted that ENGOs that communicate action-oriented messages have more followers on WeChat than those that do not, a Mann–Whitney test was utilized because it relaxes the normality assumption typical of similar parametric tests. First, the 175 ENGOs can be divided into two groups: group 1 includes 119 ENGOs (68.2%) that communicated action-oriented messages in the two-month period, and group 2 includes 56 ENGOs (31.8%) that did not communicate action-oriented messages during the period. The number of followers of the 175 ENGOs varied greatly, from 62 to 114,800 (*M* = 6073, *SD* = 11,582.9). A Mann–Whitney test revealed that Group 1 (*n* = 119) had a larger mean rank (9466.00) than Group 2 (*n* = 56) with a mean rank (2490.00). The number of followers of the ENGOs that communicated action-oriented messages was significantly higher than that of the ENGOs that did not (*U* = 1689.000, *p* < 0.05). Therefore, H3 was supported. 

## 6. Discussion

The burgeoning rise of WeChat has offered a substantial new communication platform for NGOs in China. To our knowledge, this research represents the first study that seeks to understand action-oriented communication by Chinese NGOs on WeChat. Using data on the WeChat utilization of 175 ENGOs in China, we employed both message-level and organizational-level analysis, leading to five major findings. First, although the major use of WeChat revolves around the ENGOs’ need to share information and inform their public, it is also used significantly as a tool to mobilize people to do something concrete to help achieve their missions. Second, via action-oriented messages, ENGOs motivated their publics to act in three ways: participating in various environmental activities, working with them as volunteers/interns/staff members, and donating. Our analysis suggests that WeChat serves as a powerful mobilization tool. Third, publics demonstrated higher levels of engagement when action-oriented communication was employed. Fourth, the type of action also affects levels of engagement; articles related to environmental volunteering/internships/jobs were more likely to be read. Finally, there were organizational differences in the adoption of action-oriented messages. We identified the characteristics of ENGOs adopting action-oriented communication by analyzing their locations, organizational types, and social network size. Each of the findings is discussed below.

Organizations pursuing the informing strategy try to “give sense” to their publics, which often involves passive message consumption [32]. Nevertheless, ENGOs utilize action-oriented messages to mobilize people for two-way interaction and active participation, thereby representing a higher level of commitment and engagement from publics. People who are committed to environmental protection, or at least concerned and interested in environmental issues, have the potential to be motivated to take action. It is WeChat that can connect ENGOs to such people. To be a follower of any official account of an organization on WeChat, one must request permission to be in a specific circle [14]. In other words, becoming involved in an ENGO WeChat network requires deliberate action. WeChat users intentionally choose to follow an ENGO mainly for several reasons: they are concerned about environmental problems and share their missions, or they are attracted by what it says on WeChat, and find its content interesting and useful, or they have interacted with the organization and participated in its environmental activities. Hence, the users represent a massive audience of people who have like-minded interests in the environment, and WeChat serves as a bridge between them and ENGOs. Hence, ENGOs’ action-oriented messages probably resonate with their followers, who, in turn, are more likely to be encouraged and mobilized to act. Therefore, WeChat, which is more relationship-focused, facilitates the creation of a community around an ENGO and opens up a new realm for broader civic engagement on environmental issues.

Empirical evidence reveals that ENGOs in China have been actively employing the activating strategy to inspire and motivate the public to do something for or on behalf of an organization. From an organizational perspective, this makes sense. As organizations, NGOs need to ensure a supply of the necessary resources to remain viable and fulfill their missions. These include material/financial resources and the support of various social actors, such as volunteers and clients [47]. Social media can facilitate NGOs in reaching out to larger audiences and securing a larger base from which to draw donations and acquire support. Indeed, WeChat has been the most often utilized and preferred online donation platform in China [48]. The prevalence of the use of WeChat in online donation can be attributed to two factors (as opposed to Weibo): one is that stronger social ties exist in users’ WeChat networks, and the other is that the built-in payment function of WeChat is more convenient to use [49]. Our findings show that ENGOs turn to WeChat to acquire financial resources by conducting fundraising campaigns, such as 99 Giving Day. WeChat also allows ENGOs to reach out to and engage with potential employees, volunteers, interns, partners, and people who play a role in achieving their objectives. The acquisition of these resources is probably the most tangible benefit of NGO communication on social media. From an organizational standpoint, the actions ENGOs mobilize on WeChat may be viewed as the greatest value organizations can derive from social media communication [29].

One of the major findings of this study is that action-oriented articles generate more readership than non-action-oriented articles, as reading is the primary indicator of public engagement levels on WeChat. This is probably because the public is more likely to engage with messages that are intended to foster interactions and get people involved. This finding indicates that in the digital age, new ICTs, including WeChat, could contribute to generating new paradigms of civic engagement, where the more participatory people feel, the more likely they are to respond to social media messages, and the more they are involved [50]. Users read action-oriented messages and could also share those messages with their friends on their personal networks on WeChat, who may be interested—WeChat has the potential for smaller, community-based group cohesion, and local mobilization [12]. With the higher threshold for entry, the WeChat network is centered on strong social ties that connect like-minded people [14]. Hence, sharing messages could increase the likelihood of their friends following suit in reading. The more often users share messages on WeChat, the more frequently those messages are read. Among other possible contributions WeChat might make to mobilize citizen participation in social issues, two are notable. First, WeChat, which is smartphone-based, offers the public more convenient, generalized access to a large volume of action-oriented messages in a timely manner. Second, owing to its dialogic and interactive potential, WeChat provides a medium for easy and direct communication between ENGOs and their publics. Followers can send messages directly to an official ENGO account to initiate one-to-one conservation.

This study explored which type of action reflected on ENGOs’ WeChat articles elicited more engagement from the public and found that people were more likely to read articles about environmental volunteering/internships/jobs. This finding could be better understood in the broad sociopolitical context in which NGOs, as a new social sector, have increasingly become accepted and recognized by the Chinese government and society. Recently, the government has come to realize the important role NGOs could play in social governance and pushed for the building of NGOs’ capacity to provide social services and respond to diversifying social demands [36]. NGOs are no longer viewed as “trouble makers” but instead as partners. Meanwhile, the Chinese public has gradually become familiar with NGOs that are “*gongyi zuzhi*” (public–interest organizations) working for the public good. NGOs have gained wide public recognition in China, and they have had significant popularity among certain segments of the population, such as the urban educated population [51]. The growth and mainstreaming of NGOs and their greater public visibility in China have created opportunities to gain public trust and support [52]. As noted by professors at Beijing’s top universities, their students often consider the possibility of finding jobs in NGOs, and even before graduation, students volunteer for NGOs [47]. The spirit of volunteerism is widely visible in today’s society, and it is quite common in China for college students to become involved in a volunteering program during their summer vacations [51].

The present data indicate that ENGOs in Beijing, Shanghai, Guangdong, and Yunnan or ENGOs with a larger social network were more likely to mobilize action on WeChat. One way to understand why ENGOs with such characteristics tend to communicate action-oriented messages is to see them as a response to push–pull factors. The push factors behind ENGOs’ action-oriented messages are that there is an opportunity and need to mobilize the public to act for or on behalf of an organization. With a higher level of development, capacity, and social influence, ENGOs in the four regions are more able to provide work opportunities and organize activities and projects to offer participation opportunities. Meanwhile, they have a higher potential to attract donors, volunteers, and supporters. Moreover, the pull factors include the sociodemographic factors of the regions and the size of the ENGOs’ WeChat networks. Previous research has found that levels of education and income are significantly associated with environmental awareness [53]. According to the latest census data, household income in Beijing, Shanghai, and Guangdong ranked at the top; the levels of education of Beijing and Shanghai were also top ranked, and the education levels of Guangdong ranked above average. People in these regions are more likely to have higher environmental awareness and are more likely to be mobilized to take environmental actions, thus leading to a pull factor. Similarly, our findings also indicate that ENGOs communicating action-oriented messages have many followers on WeChat. The larger the publics they have, the more potential for citizen participation ENGOs would find on WeChat, the more likely they are to mobilize action on the platform.

Regarding organizational types, service-focused ENGOs were more likely to communicate action-oriented messages than solution-focused and supportive ENGOs. This finding can be explained by examining whether they are centered on people or a specific issue. Since the 1990s, driven by serious environmental problems in China, ENGOs have often been established with the aim of implementing on-the-ground projects and programs to help address a specific environmental issue or provide support, such as funding and research for environmental protection [36]. Hence, solution-focused and supportive ENGOs have been the major types of ENGOs in China. Recently, post-materialistic values have become increasingly prevalent in the country, whereas the increasing alienation from the natural environment has been notable, which was typically true for large cities such as Beijing in an intense phase of urbanization and modernization [54]. Against this background, service-focused ENGOs have emerged that adopt programs to connect people, especially city-dwellers, with nature, cultivate environmental literacy, and promote stewardship of the earth through experiential nature education. Getting people to see, feel, touch, learn, and appreciate nature is at their core, which makes action-oriented messaging a key communicative strategy for service-focused ENGOs.

## 7. Conclusions

WeChat has been employed by Chinese ENGOs as a tool for mobilization to acquire various resources and achieve their missions. WeChat users demonstrated higher levels of engagement when ENGOs conveyed action-oriented articles, indicating that action-oriented utilization of WeChat has great potential to enable ENGOs to reach out and engage with donors, activists, supporters, partners, and the community at large. Moreover, ENGOs used WeChat to encourage and motivate people to take a range of actions from donating to volunteering to becoming involved in a variety of environmental activities. The public responded positively to all of these call-to-action messages. However, significant differences emerged between different types of action in generating public engagement. Calling to work with ENGOs as volunteers, interns, or employees elicited greater engagement, in the context of rising public environmental awareness and post-materialistic values in China. Further, ENGOs adopting action-oriented communication exhibited some characteristics in terms of organizational types, location, and WeChat network size. When an ENGO is a service-focused organization, has a large number of followers, or locates in the region of high-level ENGO development such as Beijing, it is more likely to communicate action-oriented messages on WeChat.

This study has several important theoretical and practical implications. Previous research has given little attention to the action-oriented utilization of social media. This study offers a much-needed investigation into action-oriented communication on WeChat. Existing knowledge from the fields of communication, social media, and NGO management provided important insights for starting our exploration into this topic. Our research presents an important step to go further by probing what actions are mobilized, who would mobilize their publics, and which form of action is more effective in eliciting public engagement. This study helps extend the body of knowledge on social media and stakeholder engagement in the distinctive social media landscape in non-Western contexts. Through qualitative and quantitative evidence from ENGOs’ action-oriented communicative practices, this research contributes a fresh perspective on studying the communicative practices, purposes, and prospects of NGOs. We also addressed what unique characteristics of WeChat help engender new paradigms of citizen participation in the highly Internet-connected country. It adds to the literature on the social implications of social media communication. Moreover, this study offers some practical implications for NGO practitioners. Our research illustrates the mobilizational power of WeChat messages, indicating that action-oriented content makes a significant difference in public engagement. Action mobilization seemed to be of great importance for ENGOs to acquire and sustain resources. Nevertheless, it is worth noting that not all ENGOs recognize the mobilizational potential of WeChat. Our research could help ENGOs become aware of the opportunity to reach previously untapped potential that WeChat offers to encourage citizen participation and provide insight into ways to activate the publics. An organization could go beyond merely disseminating information and informing people and focus on what it would communicate to mobilize actions to reap the benefits of action-oriented communication and effect social change.

While developing our research, we noted two limitations in the approach. First, regarding the method to investigate engagement levels, the measurement was based only on the number of reads for each article. Yet, the meaningful indicators of public engagement include both quantifiable and quantitative metrics for perception and actual action responses. To gain a more comprehensive understanding, future research may use other methods, such as surveys and interviews with ENGOs and WeChat users, to know more about how people feel concerning action-oriented messages, and whether and how users actually take action. This leads to another important question that was not addressed. This study does not examine whether the action-oriented messages have helped achieve any “real” impact on organizations, for example, whether a viewer is converted to a supporter or a donor, how many donations ENGOs acquired through the fundraising campaign initiated on WeChat, and how many volunteers are motivated by its WeChat messages. This is an area that needs to be researched in the future. 

## Figures and Tables

**Table 1 ijerph-19-03776-t001:** Actions mobilized on WeChat.

Action Types			Example	Frequency	Percentage
**Register**	**To learn**	Lecture, workshop, and forums	Household waste classification	102	15.6%
		Training courses	Environmental educator training	26	4%
	**To engage with nature**		Bird watching	118	18%
	**To partner with ENGOs**		Funding opportunities for collaborative projects	29	4.4%
	**To participate an event**	Competition and exhibition	Nature photography competition	37	5.7%
		Environmental fair	Zero waste fair	37	5.7%
		Others	Watching an environmental film	54	8.3%
**Work with ENGOs**			Recruiting volunteers/interns	146	22.3%
**Donate**			Online donation	105	16.1%
**Total**				654	100%

**Table 2 ijerph-19-03776-t002:** Frequency of ENGO WeChat articles.

Article Types	Items	Frequency	Percentage
**Action-oriented articles**	Register to participate	509	61.6%
	Work with ENGOs	146	22.3%
	Make a donation	105	16.1%
**Total**		654	100%
**Non-action-oriented articles**	ENGO activities and performance	980	53.9%
	Environmental knowledge and policy	484	26.6%
	Sharing stories of stakeholders	280	15.4%
	Holiday greetings and giving thanks	74	4.1%
**Total**		1818	100%

**Table 3 ijerph-19-03776-t003:** The number of reads of ENGOs’ articles.

Article Type	Mean	Median	SD	Minimum	Maximum
**Action-oriented articles**	555.50	315	894.670	10	9614
**Non-action-oriented articles**	354.82	136	862.890	2	16,700

## Data Availability

All the data used in this work are available on reasonable request from the corresponding author.

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
