# Peer review of "Activating beyond Informing: Action-Oriented Utilization of WeChat by Chinese Environmental NGOs"

_ijerph, 2022, doi:10.3390/ijerph19073776_

Round 1

Reviewer 1 Report

The paper presents a study that shows the way in which the adoption of social media helps engender new paradigms of citizen participation. The authors examined a dataset of 2472 WeChat articles posted by 175 environmental NGOs over a two-month period in 2019 on WeChat and identified three major types of action ENGOs mobilize on WeChat. 
My observations/recommendations regarding the paper are:
-  A more detailed description of the method of sampling and collecting data on ENGOs is important.
- A description of the iterative process of pilot testing and refining of coding rules would be required.
- Describe the method used to achieve the results in Table 2 Frequency of ENGO WeChat items.

Reviewer 2 Report

I would like to thank the authors for the opportunity to read their work. The article "Activating beyond informing action-oriented utilization of social media by Chinese Environmental NGOs," based on a content analysis of WeChat discursive material, the paper describes the impact of its uses to activate and mobilize citizens to engage environmental causes in China. The article put under eraser the gaps of existing literature. Three research questions and hypostasis are defined and well explained. In general, it is well written. An innovative topic discusses action-oriented uses of social media as a PR or an NGO communicative strategy focusing on WeChat and the Chinese context. The methodology is appropriate, and the overall result is significant. Nevertheless, there is one point that we would like to suggest improving the overall quality of the paper: a review to avoid generalization on "social media." The research is based only on WeChat. There is no comparison, for example, on the uses of Weibo. Hence, we recommend highlighting from the title to the whole article (including abstract and conclusions). This adjustment will make the paper more accurate and methodologically precise.       

Author Response

Dear Reviewer,

We truly appreciate that your comments greatly help us improve our work. Your advice has been incorporated into the revised manuscript, which is shown below. Meanwhile, all the revisions to the manuscript are marked up using the “ Track Change”.

    We are looking forward to hearing from you. Thank you!

Point 1: there is one point that we would like to suggest improving the overall quality of the paper: a review to avoid generalization on "social media." The research is based only on WeChat. There is no comparison, for example, on the uses of Weibo. Hence, we recommend highlighting from the title to the whole article (including abstract and conclusions). This adjustment will make the paper more accurate and methodologically precise.    

Response 1: We totally agree that we need to highlight WeChat from the title to the whole article, which is crucial to improve the quality of this paper and its methodological precision. Based on your advice, the title has been changed to “Activating beyond informing: Action-oriented utilization of WeChat by Chinese Environmental NGOs”. We also highlighted WeChat in the abstract, discussion, and conclusions, rather than using “social media” in general.

Reviewer 3 Report

The article raises the current topic of internet activism. It explores whether and how NGOs mobilize action on social media and the effectiveness of action-oriented messages. However, the topic is interesting, the article does not seem to be sufficiently developed. It contains a vague methodology and quite general conclusions. It is more appropriate for a journalistic article than a scientific one.

  1. The abstract is written in an incomprehensible way. Please follow the Journal instructions – the abstract should contain:

Background: Place the question addressed in a broad context and highlight the purpose of the study; 2) Methods: Describe briefly the main methods or treatments applied. Include any relevant preregistration numbers, and species and strains of any animals used. 3) Results: Summarize the article's main findings; and 4) Conclusion: Indicate the main conclusions or interpretations.

  1. Please complete the literature review, e.g. provide statistics and their source, when you write: “China is a highly internet-connected country; almost everyone has a mobile phone, and almost everyone is on WeChat”.
  2. There is a lack of methodology. In this section you provide only, how and why did you select NGO’s WeChat profiles.

In the article, you raised the questions below without providing the research methodology:

“the study aims to examine 1) How do ENGOs in China use WeChat to mobilize action? 2) How effective is the action-oriented utilization of WeChat by ENGOs? At the organizational level, it explores 3) What are the characteristics of ENGOs that adopt action-oriented communication?”

What are the indicators? How did you evaluate effectiveness?

  1. The Section Conclusion is too general. Some sentences sound like for grant application, but not a conclusion, e.g.:

“Our research presents an important step to go further by probing what actions are mobilized, who would mobilize their publics, and which form of action is more effective in eliciting public engagement. This study helps extend the body of knowledge on social media and stakeholder engagement in the distinctive social media landscape in non-Western contexts. Through qualitative and quantitative evidence from ENGOs’ action communicative practices, this research contributes a fresh perspective on studying the communicative practices, purposes, and prospects of NGOs. We also addressed what unique characteristics of WeChat help engender new paradigms of citizen participation in the highly-internet connected country. It adds to the literature on the social implications of social media communication. Moreover, this study offers some practical implications for NGO practitioners. Our research illustrates the mobilizational power of social media messages, indicating that action-oriented content makes a significant difference in public engagement.”

Reviewer 4 Report

This article studies the role of social media communications by environmental NGO's (ENGO's). More specific, it examines 2472 posts on WeChat - a platform similar to Facebook - of 175 ENGO's from 29 different provinces/autonomous regions/municipalities for a period of two months. The posts are classified to different categories.

The research questions answered by the study were:

1) What actions do ENGOs in China use WeChat to mobilize [the audience]? 
2) To what extent do ENGOs communicate action-oriented messages on WeChat?
3) How does ENGO action-oriented communication on WeChat differ by organizational type and location?

The paper is well organized. The literature review is adequate. The language does not have any flaws. The methods, although simple, serve the purpose of the study. The results seem sound. The results are not very sophisticated, but provide useful recommendations to the ENGO's. Overall, it is within the scope of the journal and I believe it will be of interest to its audience.

My only remark is that 7.5 pages, out of 20 pages overall of the article, are references. This may be due to the formatting. The formatting should be aligned with the rest of the article and if the references are still too many, they may be slightly reduced.

Author Response

Dear Reviewer,

We truly appreciate that your comments greatly help us improve our work. Your advice has been incorporated into the revised manuscript, which is shown below. Meanwhile, all the revisions to the manuscript are marked up using the “ Track Change”.

    We are looking forward to hearing from you. Thank you!

Point 1: My only remark is that 7.5 pages, out of 20 pages overall of the article, are references. This may be due to the formatting. The formatting should be aligned with the rest of the article and if the references are still too many, they may be slightly reduced.

Response 1:  We have changed the formatting of references. There are around 3 pages of references.

Round 2

Reviewer 1 Report

The paper was improved but the implemented method could have been described in more detail.

Reviewer 3 Report

Tha paper has been corrected and is accepted now.